# Towards a sector-specific $CO/CO_2$ emission ratio: Satellite-based observation of CO release from steel production in Germany

Oliver Schneising, Michael Buchwitz, Maximilian Reuter, Michael Weimer, Heinrich Bovensmann, John P. Burrows, and Hartmut Bösch

Institute of Environmental Physics (IUP), University of Bremen FB1, Bremen, Germany

**Correspondence:** O. Schneising (oliver.schneising@iup.physik.uni-bremen.de)

**Abstract.**

Global crude steel production is expected to continue to increase in the coming decades to meet the demands of the growing world population. Currently, the dominant steelmaking technology worldwide is the conventional highly $CO_2$-intensive Blast Furnace–Basic Oxygen Furnace production route (also known as the Linz–Donawitz process) using iron ore as raw material and coke as a reducing agent. As a result, large quantities of special gases that are rich in carbon monoxide (CO) are by-products of the various stages of the steelmaking process. Given the challenges associated with satellite-based estimates of carbon dioxide ($CO_2$) emissions at the scale of emitting installations due to significant background levels, co-emitted CO may serve as a valuable indicator of the carbon footprint of steel plants.

We show that regional CO release from steel production sites can be monitored from space using 5 years of measurements (2018-2022) from the TROPOspheric monitoring instrument (TROPOMI) on board the Sentinel-5 Precursor satellite benefiting from its relatively high spatial resolution and daily global coverage. We analyse all German steel plants with blast furnaces and basic oxygen furnaces and obtain associated CO emissions in the range of $50\text{-}400 \ \text{kt} \, \text{yr}^{-1}$ per site. A comparison with the respective $CO_2$ emissions on the level of emitting installations available from emissions trading data of the European Union Emissions Trading System yields a linear relationship with a sector-specific $CO/CO_2$ emission ratio for the analysed steelworks of $3.24\% \, [2.73 - 3.89; 1\sigma]$ suggesting the feasibility of using CO as a proxy for $CO_2$ emissions from comparable steel production sites. An evaluation at other steel production sites indicates that the derived $CO/CO_2$ emission ratio is also representative of other highly optimised state-of-the-art Linz–Donawitz steelworks outside Germany and that the emission ratio is potentially valuable for estimating sector-specific $CO_2$ emissions from remotely sensed CO emissions, provided that the underlying CO emission estimate is not affected by other sources.

## 1   Introduction

Carbon monoxide (CO) has a dual role as both an air pollutant and a crucial component of atmospheric chemistry. Its reactivity with the hydroxyl radical (OH) and other atmospheric constituents has a significant effect on the oxidising capacity of the atmosphere, which subsequently influences the abundances of greenhouse gases, such as methane, and the formation of secondary pollutants, including tropospheric ozone (Holloway et al., 2000). It is therefore essential to gain accurate knowledge of

the sources of CO to ensure efficient air quality control, pollution management, and an improved understanding of atmospheric composition and its impact on climate change. The atmospheric lifetime of CO is approximately 1-2 months, so that it is well suited for tracing the transport of pollutants and can be used as a proxy for co-emitted $CO_2$ (Wu et al., 2022; MacDonald et al., 2023), which is harder to observe from space than CO because $CO_2$ is well-mixed in the atmosphere resulting in small signals relative to large background levels.

Satellite missions with global coverage significantly improve our ability to analyse atmospheric CO. Instruments using measurements of shortwave infrared (SWIR) radiances to retrieve CO column abundances in the atmosphere are sensitive to the planetary boundary layer and operate during daytime when there is sufficient sunlight to illuminate the Earth's surface. Prominent instruments measuring in this spectral range include the Measurement of Pollution in the Troposphere (MOPITT) instrument (Drummond et al., 2010) on board NASA's Terra satellite (launched in 1999), the SCanning Imaging Absorption spectroMeter for Atmospheric CHartographY (SCIAMACHY) (Burrows et al., 1995; Bovensmann et al., 1999) on board ESA's ENVISAT (in operation from 2002 to 2012), and the Thermal And Near infrared Sensor for carbon Observations Fourier Transform Spectrometer 2 (TANSO-FTS-2) on board GOSAT-2 (launched in 2018) (Suto et al., 2021). The application areas for satellite data were further extended with the TROPOspheric Monitoring Instrument (TROPOMI) (Veefkind et al., 2012) onboard the Sentinel-5 Precursor satellite (launched in October 2017), which is considered a milestone for the determination of atmospheric composition, including CO, from space by combining high spatio-temporal sampling and data quality.

Major sources of CO emissions include transportation, residential heating and cooking, biomass burning, industrial processes, and natural sources like wildfires. In the broader category of industrial emissions, CO is generated as a byproduct of incomplete combustion or chemical reactions. One of the most important examples is the steel producing industry with CO emissions mainly due to the formation and release of CO-rich gases in the widely used Linz–Donawitz process of integrated steel plants: 1) Sinter gas produced in a pre-treatment process step agglomerating iron ore fines into a porous, consistent high quality burden suitable for direct use in the blast furnace, 2) blast furnace gas arising during iron ore reduction with carbon (primarily in the form of coke), and 3) converter gas resulting from conversion of molten pig iron to steel by oxidation of carbon impurities to harden the metal (Ishioka et al., 1992; Kim et al., 2016). These CO-rich by-product gases can be incinerated within the steel production route for heat supply (Backes et al., 2021) or reused in the steelmaking process to replace some of the solid carbon fuels to reduce $CO_2$ emissions substantially (He and Wang, 2017; Kildahl et al., 2023). Understanding the CO loss processes throughout the entire steelmaking workflow and their link to $CO_2$ emissions is crucial for addressing the twin challenge of environmental pollution and climate change mitigation until the planned transformation to green steel is completed. This is to be realised by the shift from the Blast Furnace–Basic Oxygen Furnace route to the Direct Reduction–Electric Arc Furnace route using pure hydrogen ($H_2$) obtained from renewable sources as the reducing agent to avoid direct emissions of non-condensable greenhouse gases almost entirely (Graupner et al., 2023).

Given that the steel industry is highly energy-intensive and one of the leading industrial contributors to global anthropogenic $CO_2$ emissions, with a share of around 7%, the rapid and systematic decarbonisation of steel production is essential to meet climate targets (International Energy Agency, 2020). With around 40 million tonnes of crude steel production, Germany is the largest steel producer in Europe and will likely exceed its sectoral $CO_2$ budget for a 1.5°C warming scenario in the 2030s at

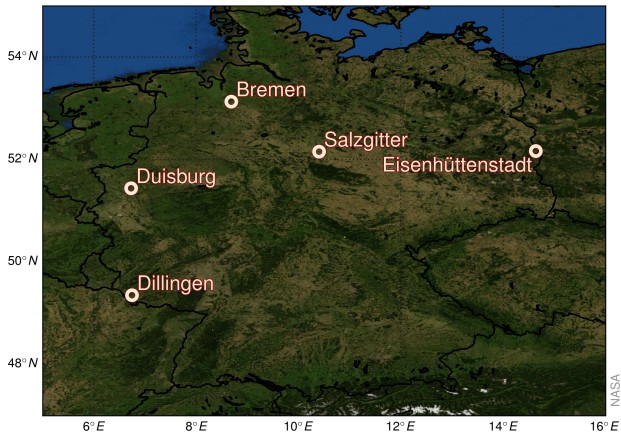

**Figure 1.** Locations of the analysed German steel production sites.

the latest (Harpprecht et al., 2022). In both Germany and worldwide, the predominant method accounting for around $70\%$ of steel production is through primary production utilising the Blast Furnace–Basic Oxygen Furnace production route.

There are several studies analysing the environmental impact of steel production sites using a bottom-up life cycle assessment (Suer et al., 2022, and references therein), but most of them focus on $CO_2$ emissions and do not quantify co-emitted CO explicitly. One exception is the study by Burchart-Korol (2013), in which all inputs and outputs of the sub-processes involved

in steelmaking are explicitly broken down for an integrated steel plant in Poland and summarised in a corresponding inventory. There are only a few studies on sector-specific CO emissions from steel production that were obtained by remote sensing. Atmospheric CO enhancements from steel production in Asia and Europe have been qualitatively analysed with TROPOMI data before (Schneising et al., 2019, 2023). Tian et al. (2022) quantify CO emission rates for industrial parks in Asia using TROPOMI data without attributing them to a specific sector. There are also some other analyses that deal with urban emissions

(Park et al., 2021; Wu et al., 2022), but these are usually a mixture of emissions from different sectors, e.g. transport, heating, and industry-related. Wu et al. (2022) derive $CO/CO_2$ ratios for cities, including two in China that are explicitly highlighted as industry- and energy-oriented with metal production sites such as aluminium or steel plants.

In this manuscript, we use TROPOMI observations and their unique features to systematically quantify the CO emissions of all German steelworks applying the Linz–Donawitz process and to determine the corresponding sector-specific $CO/CO_2$

emission ratio, which is as a first step towards the potential use of co-emitted CO as a proxy for $CO_2$ emissions from the steel industry. The locations of the analysed production sites are shown in Figure 1.

## 2   Data and methods

Sentinel-5 Precursor, launched in October 2017, operates in a sun-synchronous orbit with an equator crossing time of 13:30 local time. TROPOMI is a spaceborne nadir-viewing imaging spectrometer designed to measure solar radiation reflected by

the Earth in a push-broom configuration. With a swath width of $2600\,\text{km}$, it provides a rare combination of relatively high spatial resolution and daily global coverage. The cloud-free nadir measurements in the shortwave infrared (SWIR) offer a horizontal resolution of $5.5 \times 7\,\text{km}^2$ at nadir ($7 \times 7\,\text{km}^2$ before 6 August 2019) and are sensitive to all altitude levels, including the planetary boundary layer, making them well-suited for studying emissions from sources on the ground such as steelworks.

In this research, we use the latest version of the Weighting Function Modified DOAS (WFM-DOAS) algorithm, specifically optimised for the simultaneous retrieval of methane and carbon monoxide from TROPOMI (Schneising et al., 2019, 2023). The involved column-averaged dry air mole fractions of carbon monoxide, denoted as XCO, retrieved with the scientific TROPOMI/WFMD algorithm v1.8, are characterised by a random error (precision) of $5.1\,\text{ppb}$ and a spatio-temporal systematic error (relative accuracy) of $2.6\,\text{ppb}$ after quality filtering (Schneising et al., 2023).

The also available operational product has comparable random and systematic errors (Sha et al., 2021), but the scientific TROPOMI/WFMD product is potentially better suited for this specific application, which requires optimal near-surface sensitivity. In contrast to TROPOMI/WFMD, which is limited to clear-sky conditions, the operational product also contains scenes including mid-level clouds, i.e. cloud heights up to $5\,\text{km}$, for the case of standard quality filtering (Borsdorff et al., 2019). Although this yields a better coverage, the vertical sensitivity of the operational CO product is affected by the presence of these clouds due to cloud shielding of CO below the cloud top and scattering of electromagnetic radiation (Borsdorff et al., 2023). This complicates the interpretability of the operational CO product for applications explicitly addressing CO increases in the boundary layer and it would be necessary to assess and account for the variable vertical sensitivity of each individual sounding using the averaging kernels.

The estimation of emissions presented here relies on the daily observations provided by TROPOMI and a Gaussian integral method, also referred to as the cross-sectional flux method. To automatically process quality-filtered daily XCO retrievals for a specific source region, the estimation method of Schneising et al. (2020) is applied, which is briefly recapitulated in the following. Initially, the data given as latitude and longitude are transformed to a coordinate system rotated to the wind direction, in which the prime meridian and the equator are aligned with the source location and the zonal direction corresponds to the mean boundary layer wind direction (within a radius of $55.5\,\text{km}$ around the source and within a time window of two hours before the satellite overpass) obtained from the European Centre for Medium-Range Weather Forecasts (ECMWF) ERA5 reanalysis product (Hersbach et al., 2018). Subsequently, the transformed daily data is gridded on a $0.05° \times 0.05°$ grid, which is comparable to the horizontal nadir resolution of TROPOMI, and a mean background upwind of the source is subtracted. An example highlighting the position of the background and plume region is shown in Figure 2. Since the source location lies on the equator of the new coordinate system after the rotation, the regular latitude/longitude grid is also locally uniform in terms of distance, with $1°$ corresponding to approximately $111\,\text{km}$.

Let $E$ represent the total column enhancement (in units of mass per area) and $v$ the average wind speed within the boundary layer. To assess the emission, we use the divergence theorem and compute the flux of the vector field $E\boldsymbol{v}$ through cross sections (red lines in Figure 2) perpendicular to wind direction, which is assumed to match the plume axis after rotation of the coordinate system. Consequently, the fluxes through boundaries other than the cross sections under consideration are zero, as the unit normals of the zonal boundaries are perpendicular to wind direction and there is no flux through any upwind meridional

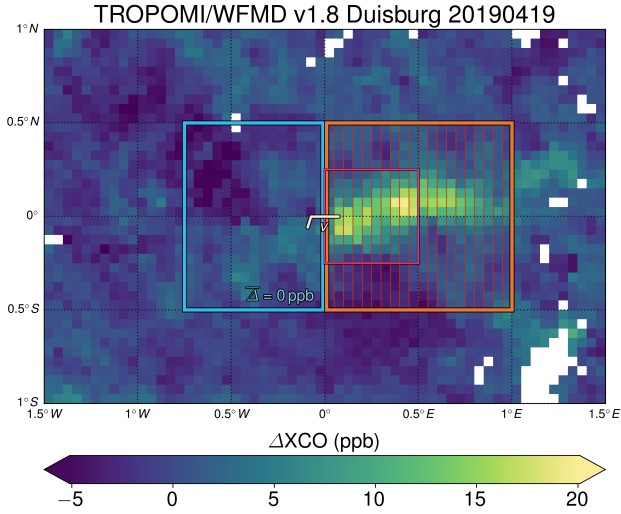

**Figure 2.** Example daily data for Duisburg to demonstrate the emission estimation process. The coordinate system has been rotated to the wind direction. The background region is highlighted in cyan, the plume region in orange, and the hot spot region in magenta. The dimensions and position of these regions remain constant for all days analysed within the rotated coordinate system. The hot spot region is only used to establish selection criteria that ensure sufficient data coverage (see main text for details) and is not used directly in the emission estimation process itself. The cross sections utilised for calculating the daily flux are indicated in red.

boundary. Thus, the emission rate is equal to the flux through the $k$th cross section for all $k$ and given by

$$\Phi_k = \int_V (\boldsymbol{\nabla} \cdot E\boldsymbol{v}) \, dV = \oint_{\partial V = S} E\boldsymbol{v} \cdot d\boldsymbol{S} = \sum_i E_{k,i} \, v \, \Delta l_i = v \, \Delta l \sum_i E_{k,i} = \frac{v \cdot \Delta l \cdot M_{\mathrm{CO}} \cdot \rho_{dry}}{N_A \cdot A_{\mathrm{CO}}} \sum_i (\Delta \mathrm{XCO})_{k,i} \tag{1}$$

Here, $\Delta l$ represents the size of a grid cell and $i$ corresponds to the meridional summation along the respective cross section given by $k$. The molar mass of carbon monoxide, $M_{\mathrm{CO}}$, is $28.01 \, \mathrm{g \, mol^{-1}}$, the Avogadro constant, $N_A$, is $6.022 \cdot 10^{23} \, \mathrm{molec \, mol^{-1}}$, and $\rho_{dry}$ (in units of molecules per area) is the mean dry air column within a radius of $111 \, \mathrm{km}$ around the source. These values are used to convert between the enhancement in XCO and the total column enhancement $E$. Additionally, $A_{\mathrm{CO}}$ is the value of the lowest layer of the averaging kernel (which is approximately 0.95 for all analysed source regions) characterising the boundary layer sensitivity of the retrieval method.

Taking the average of all cross-sectional fluxes $\Phi_k$ results in the daily flux estimate, denoted as $\Phi$. In this daily averaging procedure, only those $\Phi_k$ are considered where at least $60\%$ of the maximum possible grid cells along the cross-section are validly filled with data. If there are less than five such $\Phi_k$, this particular day will not be used any further. The corresponding total $1\sigma$-uncertainty $u_\Phi$ depends on the variability of the enhancements derived for the different cross sections, the variability of the absolute wind speed over the region, the wind history, and on the variability of the dry air column, which is determined by surface pressure (Schneising et al., 2020). The total daily uncertainty is determined by the individual uncertainty components

relative to the respective means via

$$130 \quad \left(\frac{u_\Phi}{\Phi}\right)^2 = \frac{u_{v,\text{abs}}^2 + u_{v,\text{dir}}^2}{v^2} + \left(\frac{u_{\rho_{dry}}}{\rho_{dry}}\right)^2 + \left(\frac{u_E}{\bar{E}}\right)^2 \quad\quad\quad (2)$$

with $u_{v,\text{abs}}$ being the standard deviation of all absolute boundary layer wind speed values over the selected region in the analysed time window before the satellite overpass and $u_{v,\text{dir}}$ quantifying the uncertainty in $v$ due to the maximal mean wind direction change during the considered wind history, i.e. the maximum scalar difference between wind speed and the component projected onto the mean wind direction. $u_{\rho_{dry}}$ is the standard deviation of the dry air columns within the same region used to determine the mean value (see above), $u_E$ and $\bar{E}$ are standard deviation and mean of the enhancement integrals along the different cross sections.

From the pool of available days, we select those that fulfill certain criteria (concerning data coverage, wind velocity, background scatter, meridional asymmetry, and wind history), which are listed below, for calculating the averaged long-term emission rate $\bar{\Phi}$ of the respective source region (see Schneising et al. (2020) for details). To better account for the quasi-point source character of steelworks in contrast to the larger emission regions analysed in Schneising et al. (2020), we use smaller regions according to Figure 2 (e.g. cross sections of about $111\,\text{km}$ length) and the following selection criteria optimised for XCO: at least $50\%$ of the plume, hot spot, and background regions have to be filled with data (additionally at least $20\%$ each of the northern and southern half of the background region), $v \in (1.5\,\text{m s}^{-1}, 9\,\text{m s}^{-1})$, the coverage and background distribution is required to be sufficiently uniform with respect to the equator ($n_p^N > 0.6 \cdot n_p^S$, $n_p^S > 0.6 \cdot n_p^N$, $|\bar{E}_b^N - \bar{E}_b^S| < 3\,\text{ppb}$, $\sigma(E_b) < 4\,\text{ppb}$), where $\bar{E}$ is the mean enhancement of a region, $n$ is the corresponding number of grid cells, and $\sigma$ is the standard deviation; $\cdot_{b,p}^{N,S}$ refers to the northern/southern half of the background/plume region. Additionally, days with wind direction changes exceeding $30°$ within the analysed wind history of 2 hours are also excluded. There is no criterion preventing negative meridional gradients ($\bar{E}_p < \bar{E}_b$) and associated negative emissions to ensure an unbiased mean long-term emission estimate $\bar{\Phi}$. For small emissions, inversion noise can result in negative values, whose suppression would cause an overestimation of the mean emission.

The calculation method for the $1\sigma$-uncertainty of the averaged long-term emission rate $\bar{\Phi}$ from the individual daily uncertainties has been retained exactly as described in Schneising et al. (2020); it is determined via error propagation from the individual daily uncertainties $u_\Phi$ and the number of effectively contributing days $n_{\text{eff}}$, which is smaller than the actual number of days due to expected correlation of neighbouring data points,

$$u_{\bar{\Phi}} = \frac{\sqrt{\sum_j u_{\Phi,j}^2}}{n_{\text{eff}}} \quad . \quad\quad\quad (3)$$

We assume uncorrelated data blocks with a length of one month, i.e. $n_{\text{eff}}$ is the number of months containing emission estimates contributing to the mean. Since conventional fossil fuel-based integrated steel plants are typically designed for continuous operation to provide stable and efficient production, it is assumed that there is no diurnal variation of steel production and that the TROPOMI data collected in the early afternoon represent a good approximation of the daily average. Concerning potential long-term variation in production due to changes in steel demand, it is assumed that cloud-free days are sufficiently evenly distributed over time so that the resulting temporal sampling is representative of the actual variability of steel production and

that there are thus no systematic differences in emissions between days that fulfil the selection criteria and those that do not. With decreasing temporal coverage, i.e. smaller $n_{\text{eff}}$, this representativeness potentially weakens and the uncertainty of the mean emission estimate as defined in Equation (3) increases accordingly.

The European Union Emissions Trading System (EU ETS) regulates greenhouse gas emissions of energy-intensive industries and is based on the "cap and trade" principle. The cap limits the amount of greenhouse gases allowed to be emitted by an installation and is quantified in emission allowances, which can be traded with each other. For each year, the companies operating the installations are obligated to surrender sufficient allowances to completely offset their actual emissions. Therefore, industrial installions are required to have an approved monitoring plan for monitoring and reporting annual emissions. These reported emissions data are verified annually by accredited independent verifiers (European Union, 2018). The corresponding information is available on EUETS.INFO (https://www.euets.info/, last access: 14 November 2023) providing verified annual $CO_2$ emissions and surrendered allowances on the level of emitting installations and on country level.

The $CO/CO_2$ emission ratio is determined by Theil-Sen regression of the derived CO emissions for the different sites and the respective verified $CO_2$ emissions from the production of pig iron or steel available from the emission trading data. Theil-Sen regression is a robust method, which is insensitive to outliers and influential points by using the median of the slopes of all possible lines formed by pairs of points (Sen, 1968). The corresponding confidence interval is determined by bootstrapping, fitting the Theil-Sen regressor to each resample, and analysing the resulting distribution of regression coefficients.

## 3   Results and discussion

The steel production sites in Germany that still utilise the conventional Blast Furnace–Basic Oxygen Furnace production route are located in Duisburg, Dillingen/Saar, Salzgitter, Bremen, and Eisenhüttenstadt (ordered by pig iron and steel production from high to low), which together account for approximately 70% of German steel production (Harpprecht et al., 2022). These production sites have been analysed using the cross-sectional flux method to determine CO emissions during the time period 2018-2022. Since these are the only industrial CO sources in Germany that are consistently detected in the TROPOMI data, it is justified to assume that the associated emission estimates are not affected by other sources.

### 3.1   Estimation of site-specific CO emissions

The North Rhine-Westphalian city of Duisburg has a long history as a centre of the steel industry as the region's proximity to coal and iron ore reserves made it an ideal location for steel production. The city is home to two of the most prolific steel plants in Germany in service since more than a hundred years: Integriertes Hüttenwerk Duisburg with 4 blast furnaces and Glocke Duisburg with 2 blast furnaces. The averaged multi-year CO enhancement distribution for Duisburg, which exhibits a pronounced plume structure, and the daily emission estimates are displayed in Figure 3. The associated mean CO emission estimate is determined by the daily estimates and amounts to $397 \pm 58\,\text{kt}\,\text{yr}^{-1}$.

The steel industry is one of the most important sectors of the economy in the German federal state of Saarland. Dillinger Hütte is the most significant steel producing site in the region, with production of steel dating back to the 17th century. Today,

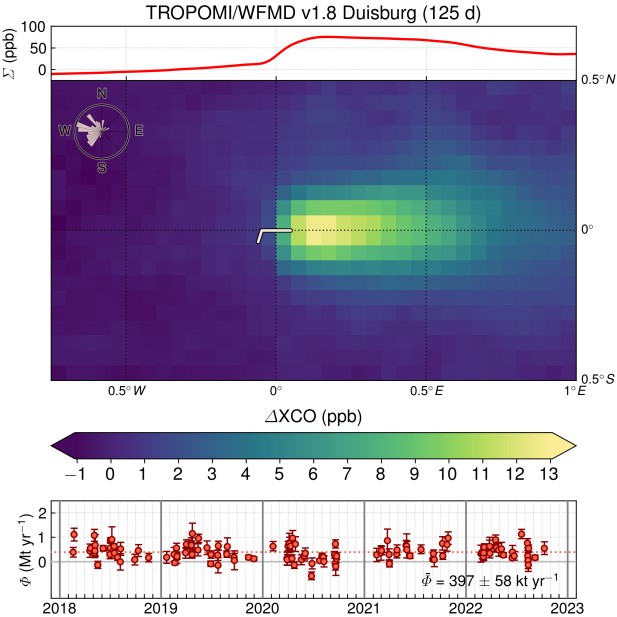

**Figure 3.** Mean CO enhancement distribution and associated integrated enhancement along the cross sections (top) as well as daily emission estimates $\Phi$ for the steel plants in Duisburg (bottom). The mean CO emission $\bar{\bar{\Phi}}$ is calculated from the daily emissions. The original daily wind directions, defined as the direction in which the wind blows, are visually represented in the wind rose overlaid in the upper left corner.

Dillingen is the only production site for pig iron in the Saarland, which is subsequently further refined into crude steel. Two blast furnaces are operated on the site. Figure 4 illustrates both the average CO enhancement distribution and the daily emission estimates for Dillingen. As in the case of Duisburg, there is a distinct plume shape visible in the long-term wind-rotated mean. The corresponding mean CO emission estimate based on the daily emissions is $157 \pm 52 \, \mathrm{kt \, yr^{-1}}$.

Salzgitter steel works is a Blast Furnace–Basic Oxygen Furnace steel plant operating in Salzgitter, Lower Saxony. The history of the site can be traced back to the interwar period in the early 20th century. During that time, an initiative aiming at establishing a self-sufficient German steel industry included the construction of a massive steel plant in Salzgitter. After shutdown and recommissioning, the site operates 3 blast furnaces today. The mean CO enhancement distribution is shown together with the daily emission estimates in Figure 5. The average CO emission is estimated to be $125 \pm 48 \, \mathrm{kt \, yr^{-1}}$.

The Bremen steel plant with 2 blast furnaces has its roots in a long history of steel production in the region. The reason for establishing steelworks at this strategic location was the access to key transportation routes, including rivers and ports, that facilitated the import of raw materials and the export of finished steel products. Over the years, the plant underwent several redesigns, mergers and expansions to meet the changing needs of the steel industry. The averaged CO enhancement distribution and the daily emission estimates are displayed in Figure 6. The associated mean CO emission estimate is $92 \pm 59 \, \mathrm{kt \, yr^{-1}}$.

The history of the Eisenhüttenstadt plant located in the German federal state of Brandenburg began in the 1950s as the Eisenhüttenkombinat in East Germany producing pig iron with 6 blast furnaces. In the 1980s, a Linz–Donawitz steelworks

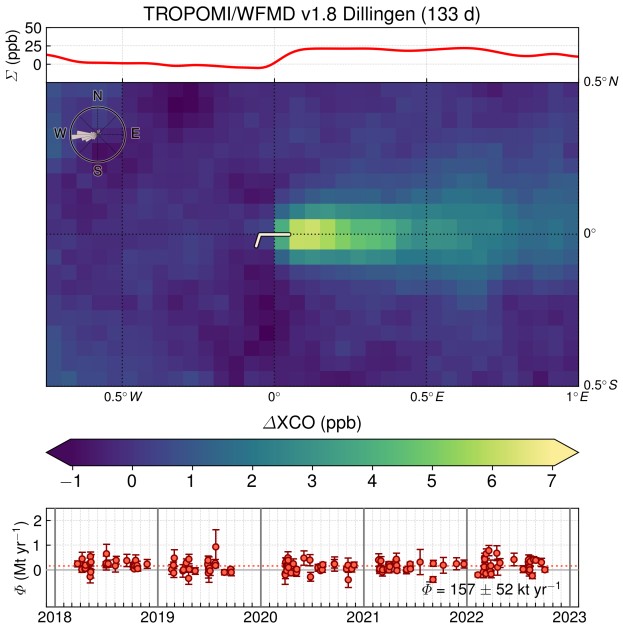

**Figure 4.** As Figure 3 but for Dillingen/Saar.

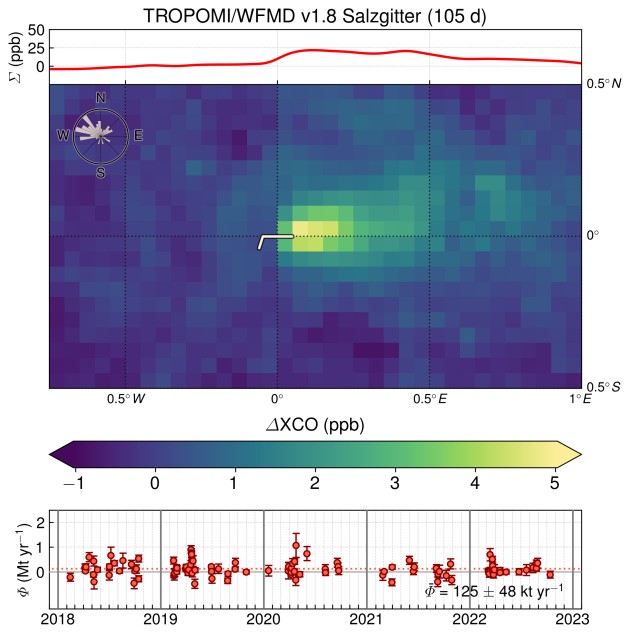

**Figure 5.** As previous figures but for Salzgitter.

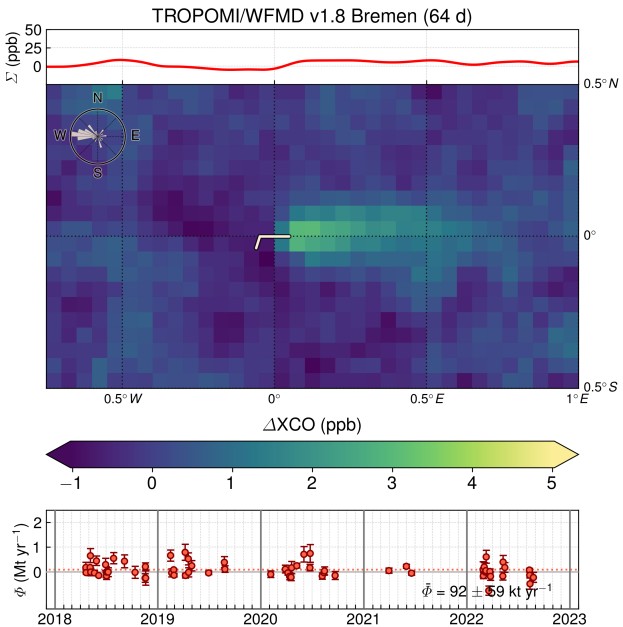

**Figure 6.** As previous figures but for Bremen.

was installed. After German reunification a new blast furnace was built, which is the only one still in operation today. Figure 7
illustrates both the average CO enhancement distribution and the daily emission estimates. The corresponding mean CO emission estimate is $48 \pm 55 \, \text{kt} \, \text{yr}^{-1}$. In contrast to the other analysed production sites, the emissions seem to be close to the detection limit because only a rather weak plume is observable in the average CO distribution. In accordance with this finding, the mean emission estimate is similar in magnitude to the corresponding uncertainty.

The partitioning of the total variance among the various uncertainty components is given according to Equation (2). For the
analysed regions, the main source of uncertainty is consistently the variability of the cross-sectional enhancements contributing on average about $90\%$ to the total variance, followed by the spatial and temporal variability of the absolute wind speed with an average contribution of about $10\%$, while the other considered uncertainty components are negligible. Concerning these prevailing sources of uncertainty, no significant site-specific differences are expected. As the temporal sampling of the daily estimates is also sufficiently evenly distributed in all cases, the uncertainty estimates are actually rather constant across
the analysed regions. Together with the consistency of the visual impression of being close to the detection limit and the approximate equality of estimated emission value and uncertainty in the case of Eisenhüttenstadt, this indicates that the derived uncertainties are generally realistic.

### 3.2   Sector-specific CO/CO$_2$ emission ratio

The European Union Emissions Trading System (EU ETS) provides verified annual $CO_2$ emissions on the level of emitting
installations. The associated mean emissions of the steel production sites for the time period analysed here (2018-2022) are

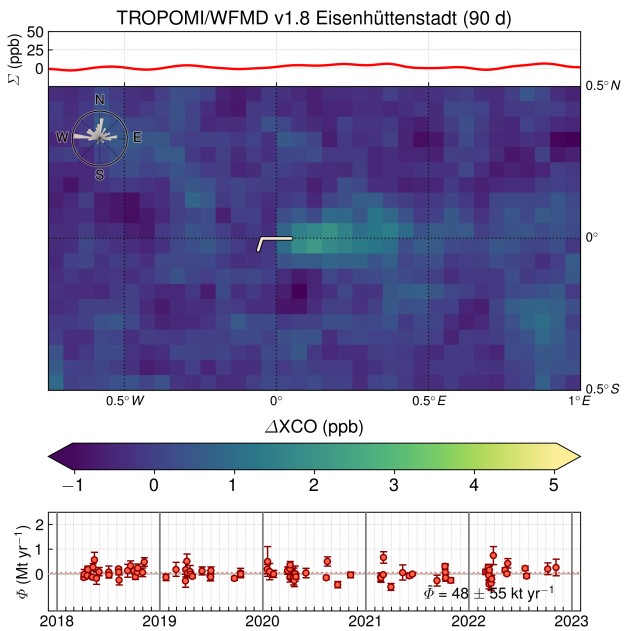

**Figure 7.** As previous figures but for Eisenhüttenstadt.

**Table 1.** Mean emissions for the time period 2018-2022. The $CO_2$ emissions from the production of pig iron or steel are obtained from the European Union Emissions Trading System (EU ETS), the CO emissions are estimated in this study as well as extracted from the Thru.de portal (providing key environmental data from industrial facilities in Germany) for comparison.

| Site | ID | Operator | Blast furnaces | EU ETS $CO_2$ emissions $(Mt\,yr^{-1})$ | This study CO emissions $(kt\,yr^{-1})$ | Thru.de CO emissions $(kt\,yr^{-1})$ |
|---|---|---|---|---|---|---|
| Duisburg | DU | Thyssenkrupp Steel Europe | 4 | 7.75 | $397 \pm 58$ | 315 |
| | | Hüttenwerke Krupp Mannesmann | 2 | 4.61 | | |
| Dillingen | DI | Dillinger Hüttenwerke | 2 | 4.15 | $157 \pm 52$ | 90 |
| Salzgitter | SZ | Salzgitter Flachstahl | 3 | 3.92 | $125 \pm 48$ | 66 |
| Bremen | HB | ArcelorMittal Bremen | 2 | 2.30 | $92 \pm 59$ | 63 |
| Eisenhüttenstadt | EH | ArcelorMittal Eisenhüttenstadt | 1 | 1.48 | $48 \pm 55$ | 40 |

summarised in Table 1. As the $CO_2$ emissions included in the EU ETS are offset by surrendering an explicit number of allowances, there are no associated uncertainties available.

Figure 8 demonstrates that CO and $CO_2$ emissions from the production of pig iron or steel are highly correlated. It is assumed that the temporal sampling of available daily CO emission estimates is representative of the actual variability of steel production and associated reported $CO_2$ emissions at the analysed sites. As the data point representing Duisburg is quite far


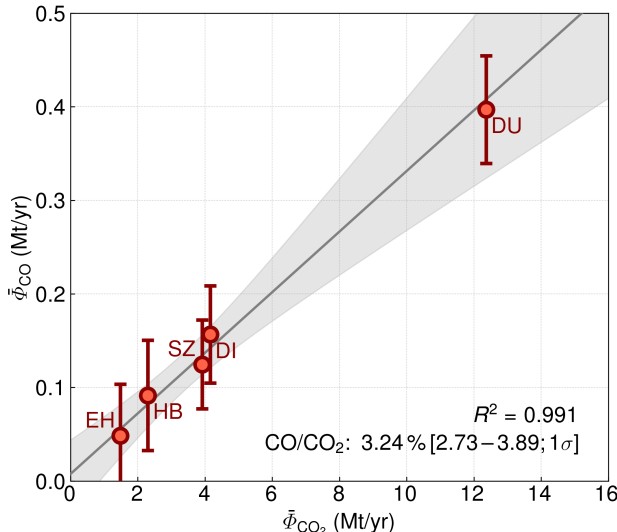

**Figure 8.** Theil-Sen regression of the derived CO emissions for the different sites and the respective $CO_2$ emissions according to the European Union Emissions Trading System (EU ETS) to determine the $CO/CO_2$ emission ratio and the associated $1\sigma$-confidence interval. The shaded region surrounding the regression line represents the prediction interval, which is determined by assessing the standard deviation of Theil-Sen regressions from multiple bootstrap samples. $R^2$ is the coefficient of determination of the prediction of the Theil-Sen regression.

away from the main cluster of data points, it may be an influential point with large impact on the results when performing an ordinary least-squares (OLS) regression, which potentially complicates the interpretation of the fitted coefficients and their uncertainties. This potential issue can be mitigated by using the robust Theil-Sen regression (Sen, 1968), which calculates the slope by taking the median of all possible slopes between individual pairs of data points. To compute a confidence interval for

the Theil-Sen regression, we use bootstrapping (plus additional random perturbation of the data points in y-direction according to the respective uncertainties of the individual CO emission estimates) to generate multiple resamples of the data and fit the Theil-Sen Regressor to each resample. This results in a set of regression coefficients from which the final regression and the associated $1\sigma$-confidence interval are derived by determining the median of slopes and intercepts as well as the 16th and 84th percentile of the distribution.

The performed robust Theil-Sen regression of CO and $CO_2$ emissions is illustrated in Figure 8. The shaded area around the regression line visually represents the prediction interval, which is determined by the standard deviation of the regressions of the multiple bootstrap samples and effectively reflects the predictive uncertainty of the model. The resulting $CO/CO_2$ emission ratio for the analysed steelworks amounts to $3.24\% \, [2.73-3.89; 1\sigma]$; the corresponding $95\%$-confidence interval (obtained from the 2.5th an 97.5th percentile of the slope distribution) is $[1.90, 5.50]$.

This sector-specific estimate can be hardly compared to global inventories such as the Emissions Database for Global Atmospheric Research (EDGAR) (Crippa et al., 2020) because the underlying sector discrimination is typically not detailed enough. For example, iron and steel production is part of the EDGAR activity "Industrial combustion and processes" that also in-

cludes sectors with significantly less CO emissions than steelworks at comparable $CO_2$ emissions. Consequently, the EDGAR $CO/CO_2$ emission ratio for the entirety of industrial processes is smaller than our sector-specific estimate for the steel industry and amounts to about $0.5\%$ for Germany. Similar representativeness issues arise when comparing to the estimated $CO/CO_2$ ratios for cities of Wu et al. (2022): their estimate for the Chinese cities with metal production is about $1\%$ (when transformed to a mass ratio) but there are also other sectors, such as energy production, with significantly smaller $CO/CO_2$ emission ratios (significant $CO_2$ emissions, but virtually no CO emissions) contributing to the derived city averages.

Another register of emissions from industrial facilities in Germany is the Thru.de portal provided by the Umweltbundesamt (https://thru.de/en/, last access: 14 November 2023) including annual loads of CO and $CO_2$ based on measurements, calculations, or estimates of the operators. The reported values are reviewed for completeness and plausibility by authorities at federal state level, but unlike the emissions trading data used to derive the $CO/CO_2$ emission ratio, no independent verification is required. Using the reported Thru.de mean releases from the German metal industry for the years 2018-2022, the associated $CO/CO_2$ emission ratio is approximately $2.0\%$. However, this value cannot be directly compared with ours either, because this activity also comprises other processes than conventional primary steel production. Deriving a sector-specific emission ratio based on Thru.de data is also not straightforward because the reported releases are not provided on the level of emitting installations (in contrast to the European Union Emissions Trading System), but on the facility level. A facility is defined as one or more installations at the same site that are run by one operator. For example, a facility in the steel industry (integrated steelworks) may include a large number of different installations (e.g. coking plant, sintering plant, blast furnace, basic oxygen furnace, rolling mill, power plant), whose combined emissions are then attributed to the so-called *main activity* of the facility, which is usually the production of pig iron or steel. While the CO emissions are actually mainly attributable to the main activity, the reported $CO_2$ emissions typically also include significant contributions from other sectors such as the energy sector. As a consequence, the facility-based Thru.de $CO/CO_2$ emission ratio underestimates the actually targeted sector-specific emission ratio, just like the other potential comparison data sets discussed so far. However, the extent of underestimation is not even consistent from site to site, but depends on which non-main activities are explicitly carried out additionally at a given facility. It is therefore not possible to obtain a representative $CO/CO_2$ emission ratio from this data set by regressing the individual CO and $CO_2$ emissions of the respective facilities. Nevertheless, the reported Thru.de CO releases can be directly compared with our emission estimates, because only the place of origin and not the exact breakdown by sector is relevant for this purpose. The corresponding values are also listed in Table 1. In general, the reported Thru.de CO releases are systematically smaller than our estimates (on average $30\%$ lower). However, the deviations in Bremen and Eisenhüttenstadt are not significant, i.e. there is agreement within the estimated $1\sigma$ uncertainty range.

The estimate that is conceptually closest to our estimate is obtained by using the outputs of the sintering plant, blast furnace, and basic oxygen furnace of the comprehensive inventory of a Polish integrated steel plant (Burchart-Korol, 2013) resulting in a $CO/CO_2$ emission ratio of $2.60\%$. This value is considerably closer to our sector-specific estimate for steel production than the other less representative estimates discussed before and within the associated $95\%$-confidence interval (even near the estimated $1\sigma$-range). This consistency further corroborates the potential feasibility of using CO as a proxy for $CO_2$ emissions from comparable steel production sites.

### 3.3 Representativeness of the derived emission ratio

A universally valid sector-specific $CO/CO_2$ emission ratio for conventional steel production would allow to convert CO emissions, e.g. estimated using satellite observations, directly to $CO_2$ emissions, which are otherwise difficult to determine by remote sensing. In this context, the question arises how representative the emission ratio determined at German steelworks is elsewhere, or in other words: What are comparable steel production sites?

The conventional Blast Furnace–Basic Oxygen Furnace steel production route has been optimised over decades to improve efficiency, reduce energy consumption, and minimise environmental impact. However, the technical limits in the optimisation of the traditional production process are gradually being reached, which means that a transformation to alternative technologies is required for the primary steel production to achieve the goal of carbon neutrality. We therefore expect that state-of-the-art Linz–Donawitz steel plants use similarly optimised raw materials and processes and that they are thus largely comparable concerning the $CO/CO_2$ emission ratio. This potentially applies to all steel plants utilising the Blast Furnace–Basic Oxygen Furnace production route in the European Union, the United States, and at least to relatively new or modernised Linz–Donawitz plants elsewhere. This assumption is evaluated by analysing additional steel production locations from this category in Slovakia, Poland, and the United States, where sector-specific $CO_2$ emission data is also available, which allows to assess whether the associated emissions are consistent with the emission ratio derived from German steelworks.

The analysed steel production sites include Košice in Slovakia with estimated CO emissions of $158 \pm 54\,\mathrm{kt\,yr^{-1}}$ and Dąbrowa Górnicza in Poland with emissions of $280 \pm 68\,\mathrm{kt\,yr^{-1}}$. In the United States, three large steelworks in northwest Indiana, which are located in close proximity to each other along the southern shore of Lake Michigan, are estimated together. The combined estimated emissions of the steel plants at Burns Harbor, Indiana Harbor, and Gary amount to $325 \pm 67\,\mathrm{kt\,yr^{-1}}$. The locations and estimated CO emissions of these steel production sites are summarised in Figure 9. Due to the location on the coast, the selection criteria concerning sufficient data coverage in the background and plume region only leave relatively few days for analysis in the case of Northwest Indiana.

The correponding $CO_2$ emissions from iron and steel production, which were extracted from the European Union Emissions Trading System (EU ETS) for the production sites in the European Union and from EPA greenhouse gas emission data (https://ghgdata.epa.gov/ghgp/, last access: 25 March 2024) for the United States, are listed in Table 2. Figure 10 compares the respective pairs of CO and $CO_2$ emissions with the previously obtained regression based on the German Linz–Donawitz steel production sites. As can be seen, the steelworks in Slovakia and Northwest Indiana are consistent with the German steel production sites concerning $CO/CO_2$ emission ratio within the estimated uncertainties, while the Polish facility does not seem to fit with the others. A more detailed analysis reveals the potential reason for this. In contrast to the other analysed sites, there are other significant CO sources in the proximity of Dąbrowa Górnicza, which likely affect the CO emission estimate and result in an overestimation of CO emissions and thus the associated emission ratio. These other CO sources include conventional steelworks in Kraków, Ostrava, and Třinec. Unlike the cases of Duisburg or Northwest Indiana, the steelworks are too far apart (about $50-100\,\mathrm{km}$ away from Dąbrowa Górnicza) to be assessed together. There are also several cement kilns (Kraków-Nowa Huta, Rudnicki, Małogoszcz, Warta) at a similar distance, which could be particularly CO-intensive if waste is

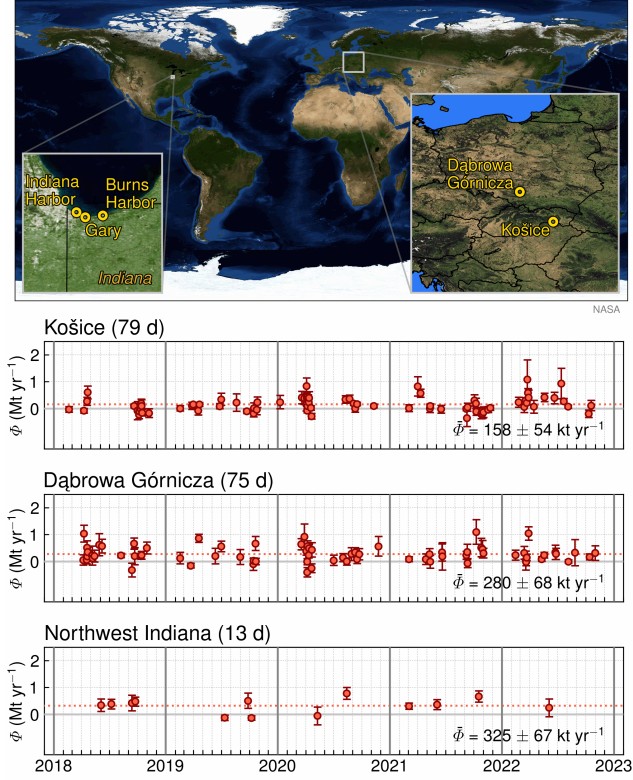

**Figure 9.** Locations and estimated CO emissions of the steel production sites used for evaluation of the emission ratio.

incinerated instead of coal in the production of cement clinker (Vilella and Arribas, 2013) without using the latest technology. As a consequence, Dąbrowa Górnicza is not an optimal target for the CO emission quantification method presented here explaining the outlier in Figure 10.

In summary, the results suggest that the sector-specific $CO/CO_2$ emission ratio determined from conventional German steel production sites is also representative of other highly optimised state-of-the-art Linz–Donawitz steelworks. However, if the ratio is to be used to derive $CO_2$ emissions from remotely sensed CO emissions, it must be ensured that the analysed facility is sufficiently isolated from other CO sources.

## 4   Conclusions

We conducted an analysis of carbon monoxide enhancements originating from German steel plants using the conventional highly $CO_2$-intensive Blast Furnace–Basic Oxygen Furnace production route. This analysis utilised daily measurements in the shortwave infrared spectral range of the TROPOMI instrument on board the Sentinel-5 Precursor satellite to estimate the CO emissions for these steelworks during the 2018-2022 period benefiting from TROPOMI's distinctive attributes, including its

**Table 2.** Mean emissions of the steel production sites used for evaluation for the time period 2018-2022. The reported $CO_2$ emissions from iron and steel production are obtained from the European Union Emissions Trading System (EU ETS) for the sites in the European Union, and from EPA GHG Emissions Data for the United States. The CO emissions are estimated in this study. The CO emission estimate for Dąbrowa Górnicza is to be treated with caution, as the analysed facility is not sufficiently isolated from other CO sources (see main text for details).

| | Site | Operator | Reported $CO_2$ emissions $(\mathrm{Mt\,yr^{-1}})$ | This study CO emissions $(\mathrm{kt\,yr^{-1}})$ |
|---|---|---|---|---|
| | Košice (Slovakia) | U.S. Steel | 5.23 | $158 \pm 54$ |
| | Dąbrowa Górnicza (Poland) | ArcelorMittal | 2.33 | $280 \pm 68$ |
| Indiana | Burns Harbor (United States) | Cleveland-Cliffs | 3.17 | |
| | Indiana Harbor (Unites States) | Cleveland-Cliffs | 2.80 | $325 \pm 67$ |
| | Gary (United States) | U.S. Steel | 2.26 | |

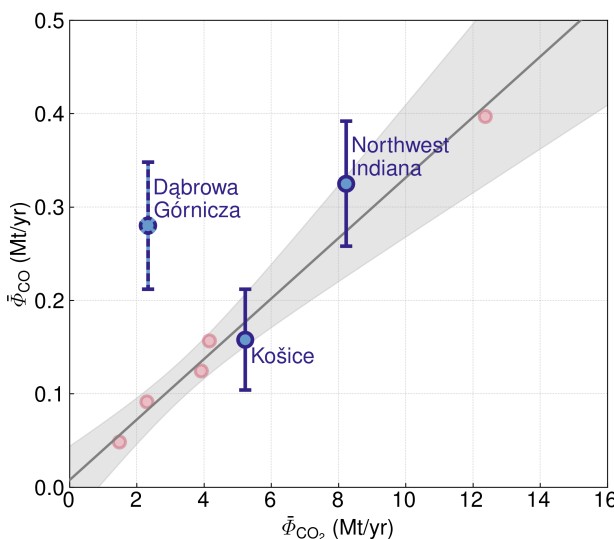

**Figure 10.** Estimated CO emissions and the respective reported $CO_2$ emissions for the different evaluation sites (blue) compared to the regression of Figure 8 used to determine the $CO/CO_2$ emission ratio based on German steel production sites (shown again in light red). The CO emission estimate for Dąbrowa Górnicza is potentially affected by nearby sources in the analysed domain (see main text for details)

high quality measurements and comprehensive spatio-temporal coverage. These qualities enable us to systematically detect and quantify sufficiently large emission sources during a single satellite overpass.

Together with the comprehensive availability of verified site-specific $CO_2$ emission data from the production of pig iron or steel available from the European Union Emissions Trading System, an estimate for the associated sector-specific $CO/CO_2$

emission ratio is derived. The high correlation of CO and $CO_2$ emissions suggests that the raw material use and the involved processes are comparable and reproducible for the steelworks studied. The focus on German steel plants has the advantage that

they are isolated from other CO sources and that the accurate reporting of $CO_2$ emissions within the framework of emissions trading enables a good calibration of the $CO/CO_2$ emission ratio, which is thus of significant value to use CO as proxy for $CO_2$ emissions from the steel industry in particular for comparable Linz–Donawitz steel plants, also in less regulated countries with less stringent reporting requirements.

If locally observed CO emissions are sufficiently sector-specific, i.e. the vast majority of CO emissions can be attributed to

a single sector, a sector-specific $CO/CO_2$ emission ratio can be used to estimate sector-specific $CO_2$ emissions from remotely sensed CO emissions. In contrast, this is difficult to achieve by means of space-based $CO_2$ measurements, because emissions from the sector in question are potentially entangled with atmospheric signals from nearby anthropogenic $CO_2$ emissions associated with other sectors as well as natural sources and sinks. For example, in the case of pig iron or steel production analysed here, the $CO_2$ emissions from steel production and electricity production are of the same order of magnitude for

a typical integrated steel plant, while the CO emissions are well dominated by steel production and the share of electricity production is negligible. Future accurate $CO_2$ satellite missions with very high spatial resolution could help to better separate emissions from neighbouring anthropogenic sources belonging to different sectors from space.

*Data availability.* The carbon monoxide data product presented in this publication is available at http://www.iup.uni-bremen.de/carbon_ghg/products/tropomi_wfmd/.

*Author contributions.* OS designed and operated the TROPOMI/WFMD satellite retrievals, performed the data analysis, interpreted the results, and wrote the paper. MB, MR, MW, HeB, JPB, HB made significant contributions to the conception of the analysis and the improvement of the manuscript. All authors discussed the results and commented on the paper.

*Competing interests.* The authors declare that they have no conflict of interest.

*Acknowledgements.* This publication contains modified Copernicus Sentinel data (2018-2022). Sentinel-5 Precursor is an ESA mission
implemented on behalf of the European Commission. The TROPOMI payload is a joint development by ESA and the Netherlands Space Office (NSO). The Sentinel-5 Precursor ground-segment development has been funded by ESA and with national contributions from The Netherlands, Germany, and Belgium. We thank the European Centre for Medium-Range Weather Forecasts (ECMWF) for providing the ERA5 reanalysis and acknowledge the use of carbon dioxide emission data from the European Union Emissions Trading System (EU ETS) (https://www.euets.info/, last access: 14 November 2023) and from EPA greenhouse gas emission data (https://ghgdata.epa.gov/ghgp/, last
access: 25 March 2024).

*Financial support.* The research leading to the presented results received funding from the European Space Agency (ESA) via the projects GHG-CCI+ and MethaneCAMP (ESA contract nos. 4000126450/19/I-NB and 4000137895/22/I-AG) and from the German ministry of education and research (BMBF) within its project ITMS via grant 01 LK2103A. The TROPOMI/WFMD retrievals presented here were performed on HPC facilities of the IUP, University of Bremen, funded under DFG/FUGG grant nos. INST 144/379-1 and INST 144/493-1. The article processing charges for this open-access publication were covered by the University of Bremen.

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
