# Peer review of "Towards a sector-specific CO/CO2 emission ratio: Satellite-based observation of CO release from steel production in Germany"

_EGUsphere, 2023_

## Author Comment (AC1)

**1 Final response to referee comments on paper egusphere-2023-2709**

We would like to thank both reviewers for their efforts in thoroughly reviewing our manuscript and for their constructive comments, which helped to further improve the paper. In the following, we provide answers and clarifications to all comments of the referees (repeated in italics).

**Anonymous Referee #1**

*Reviewer: The second last paragraph of the introduction listed related literatures that used TROPOMI CO observations for industrial emission estimation. An important difference that is not mentioned explicitly is that this study uses the WFMD product, whereas others (Park et al. 2021, Tian et al. 2022, and Wu et al., 2022) used the official S5P CO retrieval. In this paragraph or in section 2, it might be helpful to compare the WFMD and official TROPOMI CO products, e.g., in terms of algorithm design, data coverage, and precision/biases, and highlight if/why the WFMD product is advantageous for achieving the objectives of this work.*

**Authors:** We added a corresponding paragraph in Section 2: "The also available operational product has comparable random and systematic errors (Sha et al., 2021), but the scientific TROPOMI/WFMD product is potentially better suited for this specific application, which requires optimal near-surface sensitivity. In contrast to TROPOMI/WFMD, which is limited to clear-sky conditions, the operational product also contains scenes including mid-level clouds, i.e. cloud heights up to 5 km, for the case of standard quality filtering (Borsdorff et al., 2019). Although this yields a better coverage, the vertical sensitivity of the operational CO product is affected by the presence of these clouds due to cloud shielding of CO below the cloud top and scattering of electromagnetic radiation (Borsdorff et al., 2023). This complicates the interpretability of the operational CO product for applications explicitly addressing CO increases in the boundary layer and it would be necessary to assess and account for the variable vertical sensitivity of each individual sounding using the averaging kernels."

*Reviewer: Figures 2-7 appear to be after rotation, so the tick labels on the horizontal/vertical axes should not be latitude (N/S in degrees) and longitude (W/E in degrees). The spatial coordinate should be projected to uniform scales (e.g., in km) before the rotation. Also in line 90, clearly define the "region of interest" for wind averaging.*

**Authors:** The gridding to uniform latitude and longitude bins is done after the rotation. As the coordinate system is rotated such that the prime meridian and the equator are aligned with the source location, the axes in latitude and longitude after rotation in Figures 2-7 are also locally uniform in km (1° lat/lon ≈ 111 km at the equator). A corresponding clarification was added to the revised version of the manuscript: "Since the source location lies on the equator of the new coordinate system after the rotation, the regular latitude/longitude grid is also locally uniform in terms of distance, with 1° corresponding to approximately 111 km." The wind is averaged within a radius of 55.5 km around the source location. This is clarified in the revised version.

**Reviewer:** *The uncertainty calculation for the daily and long-term emission estimates are mentioned very briefly near lines 108 and 127, mostly referring to a previous work estimating methane emissions. I suggest including more details on how the uncertainties are calculated, as those are important results of this work. Specifically, the satellite only measures clear days, would there be any systematic difference in steel-producing facilities' emissions between measurable/unmeasurable days? How about the diurnal pattern of steel production, i.e., how well does TROPOMI data collected in the early afternoon represent the true daily average? Another important missing information is how the uncertainty is defined, $1\sigma$ or $2\sigma$. Both are used in the later analysis, and it's good to be consistent.*

**Authors:** In the revised version the methods to assess the uncertainties in the daily flux estimates and the averaged long-term emission rate are better explained (new Equations 2 and 3 for $u_\Phi$ and $u_{\bar{\Phi}}$). It is also clarified that both are $1\sigma$-uncertainties. As already written in Section 3.3, it is assumed that the temporal sampling of available daily CO emission estimates is representative of the actual variability of steel production and associated reported $CO_2$ emissions at the analysed sites. The underlying assumptions are also described in more detail in Section 2 of the revised version: "Since conventional fossil fuel-based integrated steel plants are typically designed for continuous operation to provide stable and efficient production, it is assumed that there is no diurnal variation of steel production and that the TROPOMI data collected in the early afternoon represent a good approximation of the daily average. Concerning potential long-term variation in production due to changes in steel demand, it is assumed that cloud-free days are sufficiently evenly distributed over time so that the resulting temporal sampling is representative of the actual variability of steel production and that there are thus no systematic differences in emissions between days that fulfil the selection criteria and those that do not. With decreasing temporal coverage this representativeness potentially weakens and the uncertainty of the mean emission estimate as defined in Equation 3 increases accordingly."

**Reviewer:** *Section 3.2, "Air quality assessment" do not seem to fit in the scope of this manuscript or contribute to the conclusion. The analysis also use very roughly estimated numbers (e.g., constant PBL height of 500 m). It is suggested to remove this section.*

**Authors:** The PBL height of 500 m was intended to be a worst-case scenario. We agree that this subsection is detached from the main conclusions of the manuscript and is therefore removed in the revised version.

**Reviewer:** *Lines 205-208: is it possible to comment on the uncertainties of x-direction, i.e., the CO2 emissions from EU ETS?*

**Authors:** As the $CO_2$ emissions included in the EU ETS are offset by surrendering an explicit number of allowances, there are no associated uncertainties available. A corresponding comment is added to Section 3 of the revised manuscript.

**Reviewer:** *Lines 246-247: it might be more accurate to conclude that the systematic low bias ($\sim20\%$, can report the exact number) relative to Thru.de exists for the estimates of all sites.*

**Authors:** The sentence is revised accordingly: "In general, the reported Thru.de CO releases are systematically smaller than our estimates (on average 30% lower). However, the deviations in Bremen and Eisenhüttenstadt are not significant, i.e. there is agreement within the

estimated $1\sigma$ uncertainty range."

**References**

Borsdorff, T., aan de Brugh, J., Schneider, A., Lorente, A., Birk, M., Wagner, G., Kivi, R., Hase, F., Feist, D. G., Sussmann, R., Rettinger, M., Wunch, D., Warneke, T., and Landgraf, J.: Improving the TROPOMI CO data product: update of the spectroscopic database and destriping of single orbits, Atmos. Meas. Tech., 12, 5443–5455, https://doi.org/10.5194/amt-12-5443-2019, 2019.

Borsdorff, T., Campos, T., Kille, N., Zarzana, K. J., Volkamer, R., and Landgraf, J.: Vertical information of CO from TROPOMI total column measurements in context of the CAMS-IFS data assimilation scheme, Atmos. Meas. Tech., 16, 3027–3038, https://doi.org/10.5194/amt-16-3027-2023, 2023.

Sha, M. K., Langerock, B., Blavier, J.-F. L., Blumenstock, T., Borsdorff, T., Buschmann, M., Dehn, A., De Mazière, M., Deutscher, N. M., Feist, D. G., García, O. E., Griffith, D. W. T., Grutter, M., Hannigan, J. W., Hase, F., Heikkinen, P., Hermans, C., Iraci, L. T., Jeseck, P., Jones, N., Kivi, R., Kumps, N., Landgraf, J., Lorente, A., Mahieu, E., Makarova, M. V., Mellqvist, J., Metzger, J.-M., Morino, I., Nagahama, T., Notholt, J., Ohyama, H., Ortega, I., Palm, M., Petri, C., Pollard, D. F., Rettinger, M., Robinson, J., Roche, S., Roehl, C. M., Röhling, A. N., Rousogenous, C., Schneider, M., Shiomi, K., Smale, D., Stremme, W., Strong, K., Sussmann, R., Té, Y., Uchino, O., Velazco, V. A., Vigouroux, C., Vrekoussis, M., Wang, P., Warneke, T., Wizenberg, T., Wunch, D., Yamanouchi, S., Yang, Y., and Zhou, M.: Validation of methane and carbon monoxide from Sentinel-5 Precursor using TCCON and NDACC-IRWG stations, Atmos. Meas. Tech., 14, 6249–6304, https://doi.org/10.5194/amt-14-6249-2021, 2021.

**Anonymous Referee #2**

**Main comments**

**Reviewer:** *While I understand the point of not including information on how the uncertainty is calculated and referring to the method paper by Schneising et al. (2020b), I think it requires some more discussion once the results are shown and, for example, on page 8 line 178, it is stated that "the derived uncertainties are generally realistic". The uncertainty values are rather constant, but aren't there different characteristics from the different areas causing different challenges to the emission rate estimation process that would be reflected in the uncertainty estimate?*

**Authors:** We decided to add the formulas how the uncertainties of the daily flux estimates and the averaged long-term emission rate are calculated to the revised version of the manuscript because this makes it easier to discuss the resulting uncertainties. The daily uncertainties depend on individual uncertainty components quantifying the impact of the variability of the enhancements derived for the different cross sections, the variability of the absolute wind speed and wind direction, and on the variability of the dry air column (new Equation 2). The main source of uncertainty for the analysed regions is consistently the variability of the cross-sectional enhancements contributing on average about 90% to the total variance, followed by the spatial and temporal variability of the absolute wind speed contributing on average about 10% to the variance. The other considered uncertainty components are negligible. Concerning these prevailing sources of uncertainty, no significant site-specific differences are expected for the analysed locations. As the temporal sampling of the daily estimates is also sufficiently evenly distributed in all cases, the uncertainty estimates are actually rather constant across the analysed regions. Together with the consistency of the visual impression of being close to the detection limit and the approximate equality of estimated emission value and uncertainty in the case of Eisenhüttenstadt, this indicates that the derived uncertainties are generally realistic. This discussion is added to the revised version to make the argument clearer.

**Reviewer:** *Are CO emissions assumed to be a continuous source? How does this assumption affect the estimated emissions? Authors mention using wind history of 2 hours to apply filtering criteria, but what if the emissions have been happening for more than 2 hours before the satellite overpass? Also, the plume may have travelled further than the selected box downwind.*

**Authors:** It is assumed that there is no diurnal variation of steel production and that the TROPOMI data collected in the early afternoon represent a good approximation of the true daily average, because conventional fossil fuel-based integrated steel plants are typically designed for continuous operation to provide stable and efficient production. The production may change in the long term, e.g. to adapt to changes in steel demand, but it is assumed that the temporal sampling of available daily emission estimates is representative of the actual variability of steel production. If the sampling becomes more uneven, the estimated uncertainty also increases (new Equation 3). A corresponding discussion of the underlying assumptions is added to Section 2: "Since conventional fossil fuel-based integrated steel plants are typically designed for continuous operation to provide stable and efficient production, it is assumed that there is no diurnal variation of steel production and that the TROPOMI data collected in the early afternoon represent a good approximation of the daily average.

Concerning potential long-term variation in production due to changes in steel demand, it is assumed that cloud-free days are sufficiently evenly distributed over time so that the resulting temporal sampling is representative of the actual variability of steel production and that there are thus no systematic differences in emissions between days that fulfil the selection criteria and those that do not. With decreasing temporal coverage this representativeness potentially weakens and the uncertainty of the mean emission estimate as defined in Equation 3 increases accordingly."

As the emissions are constant over the day, they also occur more than 2 hours before the satellite overpass. However, the plume near the source is determined by the short-term wind history. Therefore, a time window of two hours was chosen. It is not a problem if the plume leaves the box downwind as long as we have enough cross sections to obtain a reliable estimate (as stated in the manuscipt, we require at least 5 cross sections here). Each cross-section provides its own emission estimate and their average provides the daily emission estimate. More available cross-sections typically reduce the daily uncertainty but it is not a conceptual problem if you miss cross-sections, neither inside nor outside the box.

**Reviewer:** *The day-to-day variability of emissions is not discussed. This is fine as yearly CO emissions is the value that is used for the ratio, but also because this reason I fail to understand why instead of computing annual average from daily emissions, the method has not been applied to the annual average of CO concentrations. Are those two estimates (annual from daily and annual) consistent with each other?*

**Authors:** Calculating mean emissions from daily estimates has the advantage that error propagation of the daily uncertainties can be used. Moreover, it provides more accurate results if you first analyse the increases associated with the individual wind directly on a daily basis before averaging because you potentially lose information if you compute the emissions based on averaged wind and averaged enhancements, e.g. in case of large variability of daily wind speeds or uneven daily spatial sampling. However, quite consistent results within the estimated uncertainties are found here due to the rather favourable conditions in this respect (e.g. in the case of Duisburg, $419\,\mathrm{kt\,yr^{-1}}$ for emission from averages instead of $397\,\mathrm{kt\,yr^{-1}}$ for the average of emissions used here). Nevertheless, averaging daily emissions (if possible) is generally the better choice than averaging wind and enhancements first.

**Reviewer:** *Air quality assessment: assuming a boundary layer height for all days for all locations needs to be proven valid, especially the wide variety of emission ranges found depending on the location.*

**Authors:** The PBL height of $500\,\mathrm{m}$ was intended to be a worst-case scenario. Since this subsection is detached from the main conclusions of the manuscript, it is removed in the revised version as suggested by Referee #1.

**Reviewer:** *The possibility of having other CO sources in the domain should be further discussed. How does it affect the results as you compare facility level CO2 emissions to domain-wide CO emissions?*

**Authors:** In order to calculate a sound sector-based $CO/CO_2$ emission ratio or to derive sector-specific $CO_2$ emissions from estimated CO emissions using the previously calibrated emission ratio, there must be no other significant sources of CO in the analysed region than the

analysed steelworks. This is mentioned in the abstract, in the conclusions, and also discussed in the context of the question of applicability outside Germany in the revised version (new subsection 3.3). In the case of the analysed German steel plants, there are no other interfering CO sources and the conventional steel plants are the only industrial CO sources that are consistently detected in the TROPOMI data across the country. A corresponding statement is added at the beginning of section 3. Small fires also occur from time to time, but none of them interfered with the sources analysed here. It is also stated in the conclusions that the isolation from other CO sources is one of the advantages of using German steel plants to calibrate the sector-based $CO/CO_2$ emission ratio. In the case of a Polish steel plant additionally discussed in the new subsection 3.3, the situation arises that other CO sources are in the proximity affecting the applicability of the method.

**Reviewer:** *Given the nature of CO2 emissions data used in this study, does that imply that this study only applies to Germany? Are there other areas where this type of granular and process specific CO2 data is available? This should be further discussed in the conclusions section.*

**Authors:** The idea is to calibrate the emission ratio using Germany as a suitable target where the steel works are sufficiently isolated from other significant CO sources and $CO_2$ emission data is available at emitting installation level, and then ideally use this ratio to estimate $CO_2$ emissions from satellite-based CO emission estimates elsewhere. This is clarified in the abstract and in the conclusions of the revised version. We also added a new subsection discussing the representativeness of the derived emission ratio for other steel plants outside Germany. To this end, we analyse 3 other facilities in Slovakia, Poland, and the U.S., where sector-specific $CO_2$ emission data is also available, and assess whether the associated emissions are consistent with the emission ratio derived from German steelworks.

**Reviewer:** *In the introduction or in the conclusion section, I would recommend highlighting the implications and advantage of having a CO/CO2 ratio. Why one would be interested in such a number?*

**Authors:** Along the lines of the previous answer, this is better clarified in the abstract and in the conclusions of the revised version.

**Specific/Technical comments**

**Reviewer:** *First paragraph page 2: explain further the time ranges in operation of these satellites to put the reader in context. Also, is there any relevant work with these satellites from the CO steel production perspective?*

**Authors:** The operation time ranges of the instruments are included in the introduction. Relevant work is referenced at the end of the section. We are not aware of significant contributions from the CO steel production perspective before the launch of Sentinel-5P.

**Reviewer:** *Page 2, line 39: how important it is in terms of CO emissions of the industry as compared to other sources?*

**Authors:** Steel production is one of the most important examples of industrial CO emissions.

This is highlighted more strongly in the revised version. In Germany, conventional steel plants are the only industrial CO sources that are consistently detected in the TROPOMI data. A corresponding statement is added at the beginning of section 3.

**Reviewer:** *Lines 46-51: long sentence, split into two for readability.*

**Authors:** The sentence has been split in the revised version.

**Reviewer:** *Line 250: "than the other less representative estimates": please explain which you are referring to.*

**Authors:** The statement refers to the estimates discussed before. This is clarified in the revised version.